# A theory that predicts behaviors of disordered cytoskeletal networks

Julio M Belmonte[1,2] iD, Maria Leptin[1] & François Nédélec[2,*] iD

## Abstract

Morphogenesis in animal tissues is largely driven by actomyosin networks, through tensions generated by an active contractile process. Although the network components and their properties are known, and networks can be reconstituted *in vitro*, the requirements for contractility are still poorly understood. Here, we describe a theory that predicts whether an isotropic network will contract, expand, or conserve its dimensions. This analytical theory correctly predicts the behavior of simulated networks, consisting of filaments with varying combinations of connectors, and reveals conditions under which networks of rigid filaments are either contractile or expansile. Our results suggest that pulsatility is an intrinsic behavior of contractile networks if the filaments are not stable but turn over. The theory offers a unifying framework to think about mechanisms of contractions or expansion. It provides the foundation for studying a broad range of processes involving cytoskeletal networks and a basis for designing synthetic networks.

**Keywords** actin; active gel; cell cortex; contractility; morphogenesis
**Subject Categories** Cell Adhesion, Polarity & Cytoskeleton; Quantitative Biology & Dynamical Systems
**Mol Syst Biol. (2017) 13: 941**

## Introduction

Networks of cytoskeletal filaments display a variety of behaviors. A decisive feature for the physiological role of networks is whether they contract or expand. For instance, actomyosin cortices can contract, and the tensions thus created determine the morphology of animal cells (Salbreux *et al*, 2012; Maître *et al*, 2016). Conversely, the mitotic spindle at anaphase is a network of microtubules that extends to segregate the chromosomes. Such behaviors are essential, but we still lack an intuitive understanding of how they come about, as it is difficult to extrapolate between the microscopic level, where filaments are moved by molecular motors and restrained by crosslinking elements, and the level of the entire system.

Cytoskeletal filaments and many of their associated factors are well characterized biochemically. With sufficient knowledge of the relevant properties of the components of a particular network, it should be possible to predict the network behavior. Traditional approaches were particularly successful in predicting passive systems composed of reticulated polymers (Wolff & Kroy, 2012), and more recent developments in active gel theories address networks containing molecular motors (Prost *et al*, 2015). These latter theories however cannot explain the contractile or expansile nature of the network, as it arises from microscopic interactions that are not represented in these theories. To understand why contractility occurs, one must describe the system at higher resolution and consider motors and filaments individually (Kruse & Jülicher, 2000; Liverpool & Marchetti, 2003; Liverpool *et al*, 2009). Small networks can also be studied with computer simulations (Mendes Pinto *et al*, 2012; Stachowiak *et al*, 2014; Oelz *et al*, 2015; Ennomani *et al*, 2016; Hiraiwa & Salbreux, 2016), but we lack a simpler approach that can make rapid predictions purely based on analytical deduction. Such a theoretical framework would be particularly valuable to classify the different behaviors that are seen experimentally.

In search for such a general theory, we chose initially to concentrate on the major factor determining contraction of networks, that is the force created by molecular motors, although we recognized that filament shortening could also lead to contractility (Backouche *et al*, 2006; Mendes Pinto *et al*, 2012; Oelz *et al*, 2015). *In vitro* experiments have shown that contractility can arise with stabilized filaments. In such experiments, the filaments are initially distributed randomly, and molecular motors or crosslinkers added to the mixture make random connections between neighboring filaments. The active motions of molecular motors then drive network evolution. With microtubules and kinesin oligomers, static patterns such as asters (Nedelec *et al*, 1997; Köhler *et al*, 2011) or dynamic beating patterns (Takiguchi, 1991; Katoh *et al*, 1998; Sanchez *et al*, 2011; Thoresen *et al*, 2011) can arise. While radial (Backouche *et al*, 2006) and other patterns (Köhler *et al*, 2011) were also observed with actin, F-actin networks activated with myosin are predominantly contractile, as demonstrated in various geometries: bundles (Takiguchi, 1991; Katoh *et al*, 1998; Thoresen *et al*, 2011), rings (Reymann *et al*, 2012), planar networks (Murrell & Gardel, 2012), spherical cortices (Carvalho *et al*, 2013; Vogel *et al*, 2013; Shah *et al*, 2014), or 3D networks (Bendix *et al*, 2008; Koenderink

1   Directors's Research/Developmental Biology Unit, European Molecular Biology Laboratory, Heidelberg, Germany
2   Cell Biology and Biophysics Unit, European Molecular Biology Laboratory, Heidelberg, Germany
    *Corresponding author. Tel: +49 6221 387 597; E-mail: nedelec@embl.de

*et al*, 2009). Microtubule networks with NCD or dynein motors are also contractile (Surrey *et al*, 2001; Foster *et al*, 2015). Several interesting mechanisms of contraction have been proposed and reviewed recently (Murrell *et al*, 2015), but each of these applies only to a particular system for which it explains the behavior. We propose here a general theory that can be applied to both microtubule and actin systems. We also show that contractile systems become pulsatile if filament turnover is introduced in the model.

## Results

### A simple theory to predict the behavior of random networks

Let us consider a disorganized set of filaments connected by active and passive "connectors" made of two functional subunits (Fig 1A and B). Examples for passive connectors are crosslinkers such as Ase1, Plastin, alpha-Actinin, or Filamin, whereas active connectors are oligomeric motors such as myosin minifilaments, dynein complexes, bifunctional motors such as kinesin-5 or myosin VI, that are able to connect two filaments at the same time. By walking along filaments, bridging motors move the filaments relative to each other and change the network. It is however not obvious *a priori* how the sum of their local effects will influence the overall shape and size of the network. A computer can be used to simulate the dynamics of a network, but because all biochemical parameters must be specified in a simulation, only a finite set of conditions can be tested. We present here an analytical theory that overcomes this limitation. Active networks have been previously analyzed (Nedelec *et al*, 1997; Liverpool & Marchetti, 2003; Ziebert *et al*, 2007; Gowrishankar *et al*, 2012; Lenz, 2014) by considering pairs of filaments with one active connector (Fig 1C). This approach is valid for sparsely connected networks in which only a few motors are active, but physiological networks must be well connected to exert force. In other words, the network should be elastically percolated, and there must exist continuous paths through which tension can be transmitted between any pair of distant points (Dasanayake *et al*, 2011). Specifically, we assumed that filaments are connected to at least two other filaments of the network. Focusing on one of these filaments (Fig 1D), we see that the section of filament between two connectors acts as an elementary mechanical bridge between two points of the network. If the connectors are immobile, or if they

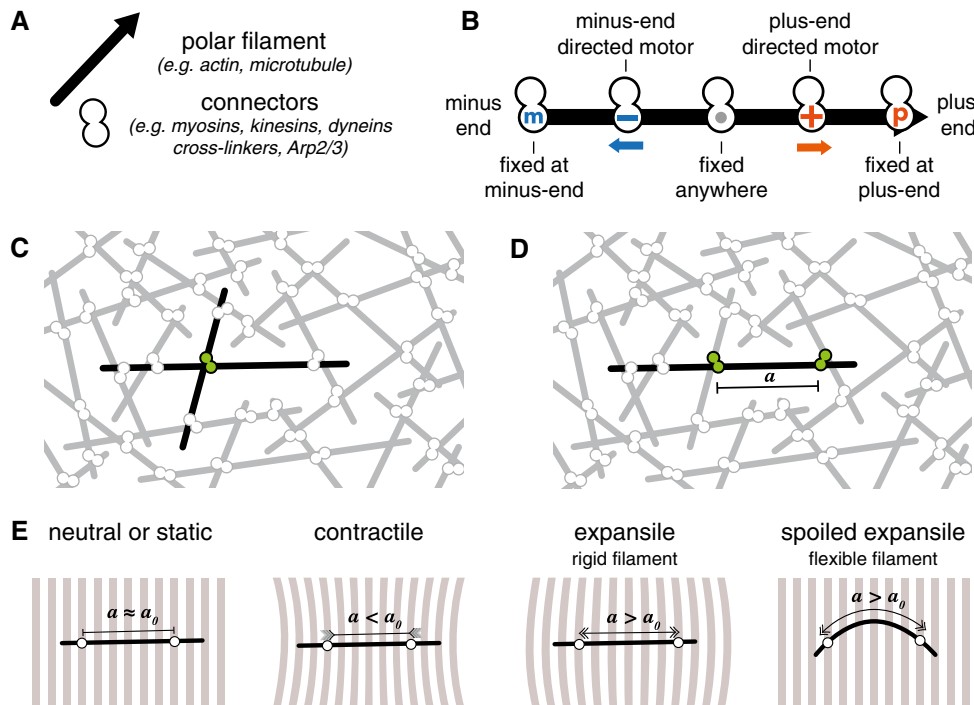

**Figure 1. Elements of active networks.**

A   Networks are composed of polar filaments that may bend, and connectors containing two subunits through which they can bridge two nearby filaments.

B   Subunits may be minus-end- or plus-end-directed motors that can bind anywhere to a filament, or binders that can bind to any location along a filament, or end binders that attach only at the minus or the plus ends of filaments.

C, D   To predict the behavior of a network, previous theories have considered a pair of filaments with a single connector between them, while the theory presented here is based on the effects that two connectors bound to a single filament have on the rest of the network.

E   Pairs of connectors may generate local stress in the network depending on how the subunits move relative to one another on the filament. If the initial distance $a_0$ between the subunits is maintained, the network does not deform. This occurs if the connectors do not move (in static configurations) or if they move in the same direction at the same speed (in neutral configuration). Local contraction is expected for contractile configuration in which the connectors move toward each other ($a < a_0$) and expansion may occur for extensile configuration where the connectors move apart ($a > a_0$). If the filament is flexible, however, the expansile stress can be reduced if the filament buckles.

both move in the same direction at the same speed, their distance remains constant, the section of the filament between them does not change in length, and the bridge is neutral (Fig 1E). By contrast, if the two connectors move toward each other, the bridge exerts a contractile stress, whereas if they move apart, this produces an expansile stress (Fig 1E).

To predict whether the whole network will contract or expand, we sum up the effects of all elementary mechanical bridges in the network. To do this, we first list all the possible configurations for two connector subunits bound to a filament (Box 1). For each configuration, we then determine the distance between the two connectors measured along the filament $a_i$, and how this distance changes over time: $v_i = \frac{da_i}{dt}$. To calculate $v_i$, we only consider the nature of the bound subunits of the connectors and thus use the unloaded speed of the motors, rather than their actual speed. We then sum all the contributions (Box 1B), taking into account the probability $p_i$ of each configuration to occur, which can be calculated from the concentrations of components in the system, the binding and unbinding rates of the subunits, and other characteristics of the network (see Appendix Supplementary Methods). We also distinguish the case where the filaments are rigid and can support expansile stress from the case where the filaments are flexible such that they buckle under compression (Box 1B). In the latter case, filament buckling spoils part or all of the expansile forces (Fig 1E), and

we thus discard the contribution of these expansile configurations. The ratio between two sums calculated over all configurations predicts the network behavior (Box 1C) and can be calculated algebraically.

### Actomyosin networks with motors and crosslinkers

To test and develop the theory, we first applied it to a much studied model of cytoskeletal activity, that of actomyosin contraction, which has also been reconstituted *in vitro* (Takiguchi, 1991; Katoh *et al*, 1998; Mizuno *et al*, 2007; Koenderink *et al*, 2009; Thoresen *et al*, 2011; Murrell & Gardel, 2012; Reymann *et al*, 2012; Carvalho *et al*, 2013; Vogel *et al*, 2013; Shah *et al*, 2014). Actomyosin networks consist of stabilized F-actin filaments and two types of connectors: bifunctional motors moving at speed $v$ and passive crosslinkers (Fig 2A). Bifunctional motors are connectors composed of two motor subunits that bind anywhere on the filament and move toward one end of the filament, in this case the plus-end, at a load-dependent velocity. The crosslinker is composed of two identical subunits that may bind anywhere on the filaments, and that remain immobile until they detach. There are four possible ways to arrange the two types of connectors on a filament (Fig 2A). Their likelihood depends on $P_M$ and $P_C$, the probability of one or more motors, and the probability of one or

---

**Box 1:  Analytical prediction of contraction/expansion rate.**

### A  List all possible configurations

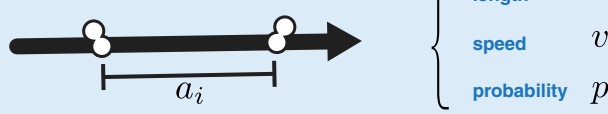

$$\begin{cases} \text{length} & a_i \\ \text{speed} & v_i = \dfrac{da_i}{dt} \\ \text{probability} & p_i \end{cases}$$

### B  Combine all contributions

$$\chi = \frac{\sum_i \Phi(a_i, v_i) p_i v_i}{\sum_i p_i a_i}$$

rigid filaments:        $\Phi = 1$

semi-flexible filaments:   $\Phi = \begin{cases} 0 & \text{if } v_i > 0 \text{ and } a_i > b \\ 1 & \text{otherwise} \end{cases}$

flexible filaments:      $\Phi = \begin{cases} 0 & \text{if } v_i > 0 \\ 1 & \text{otherwise} \end{cases}$

### C  Predicted outcome

**1D network of length L**

$$\frac{1}{L}\frac{dL}{dt} \approx \chi$$

**2D network of surface S**

$$\frac{1}{S}\frac{dS}{dt} \approx 2\chi$$

**3D network of volume V**

$$\frac{1}{V}\frac{dV}{dt} \approx 3\chi$$

The behavior of a disorganized network of filaments can be predicted analytically following a three-step procedure. (A) A list of all possible configurations involving one filament and two connectors is compiled. For each configuration, the separation $a_i$ between the connectors, the speed $v_i$ at which the they move in relation to one another, and the likelihood $p_i$ of finding the configuration within the network are noted. (B) These quantities are combined into a scalar $\chi$, using a function $\Phi$, depending on the nature of the filaments. For rigid filaments that do not buckle, all contributions are added ($\Phi = 1$). For flexible filaments that buckle readily under compression, only contractile configurations ($v_i < 0$) are retained. For a network made of semi-flexible filaments, expansile configurations above the buckling threshold $b$ are discarded. (C) The scalar $\chi$ predicts the contraction rate of the network, depending on its dimensionality, as indicated. The sign of $\chi$ indicates if the network is contractile ($\chi < 0$) or expansile ($\chi > 0$).

---

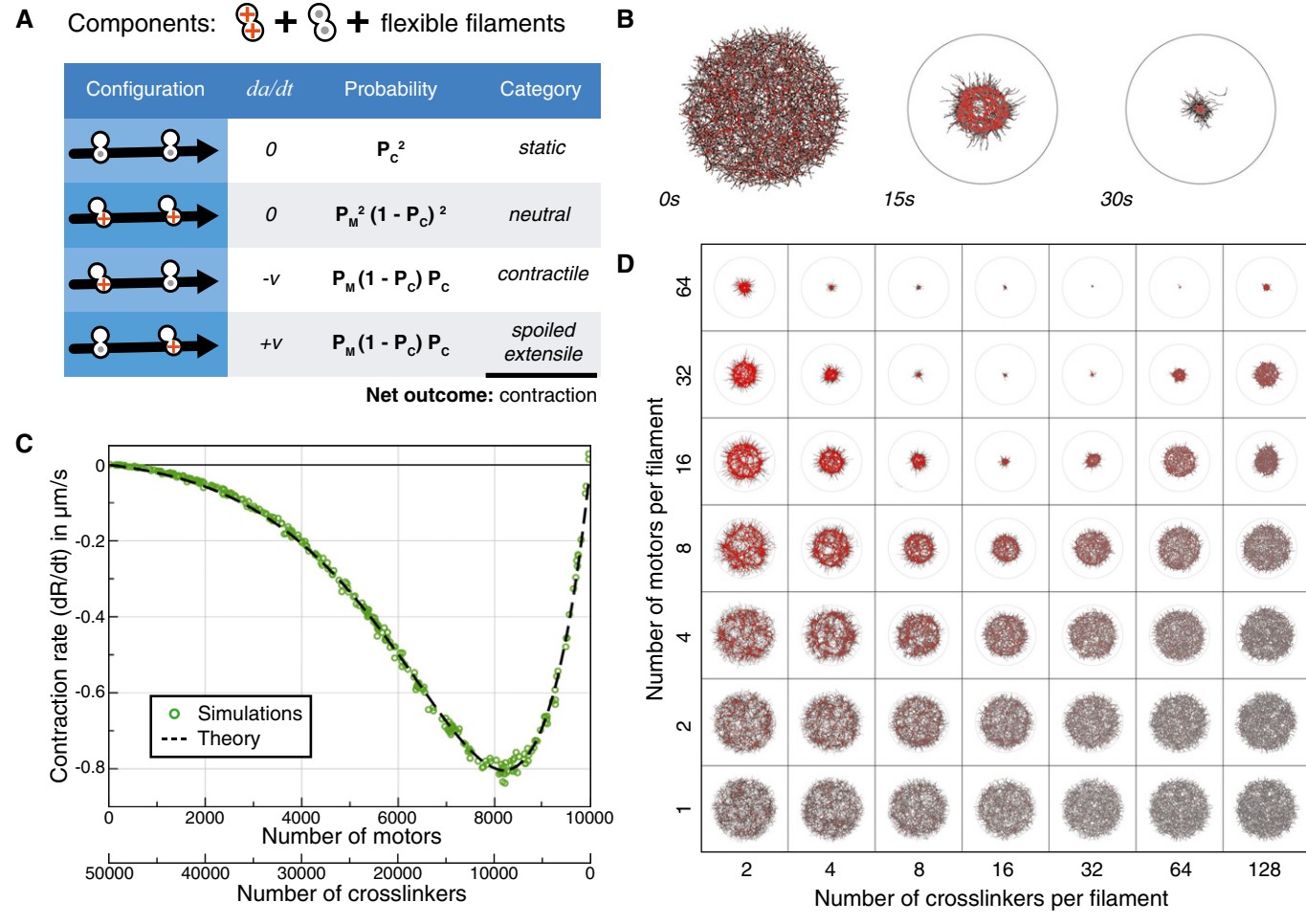

**Figure 2.  Predictions and simulations for actin-like networks of flexible filaments.**

A   A system composed of flexible filaments and two types of connectors: crosslinkers and bifunctional motors. The table lists the four possible configurations for two connectors bound to a filament, the relative movement of the connectors ($\frac{da}{dt}$), and the likelihood and the mechanical nature of each configuration. The likelihoods are combinations of $P_M$ and $P_C$, which are the probabilities of having at least one motor or one crosslinker at an intersection of two filaments (see Appendix Supplementary Methods D).

B   The evolution of a simulated random network composed of 1,500 flexible filaments (bending rigidity = 0.01 pN μm²) and 12,000 connectors of each type, distributed over a circular area of radius 15 μm.

C   The contraction rate of a simulated network as a function of the ratio of crosslinkers to motors, with the total number of connectors kept constant. Each symbol indicates the result of one simulation. The broken line indicates the analytical prediction made by the theory (see Appendix Supplementary Methods D). No contraction occurs without crosslinkers or without motors, and the maximal contractile rate is obtained here for 8,000 motors and 10,000 crosslinkers.

D   Snapshots at $t$ = 10 s of networks similar to (B) containing varying numbers of motors (vertical axis) and crosslinkers (horizontal axis).

Source data are available online for this figure.

more crosslinkers being bound at an intersection of filaments, respectively (Appendix Supplementary Methods D). The configuration with two crosslinkers is passive. The one with two motors is neutral, because the motors move in synchrony and retain their distance. The other configurations involve a motor and a crosslinker (Fig 2A). They are active with opposite outcomes. In one, the motor and the crosslinker approach each other at speed $-v$, and in the other, they move apart at speed $v$. They have an equal likelihood that is proportional to $P_M P_C (1-P_C)$, reflecting that one of the crossings should have at least one motor and no crosslinkers, with a likelihood $P_M (1-P_C)$, while the second crossing should have at least a crosslinker, with or without motors, carrying a

likelihood $P_C$. The net sum over the effects of all configurations in this example is null, and this predicts that a system made of rigid filaments that remain straight should neither contract nor expand. Contractile and expansile configurations cancel each other out, as found previously in the case where only motors were considered (Kruse & Jülicher, 2000). If the filament buckles, however, the expansile configuration will not be able to drive network expansion (Fig 1E, last panel). Whether a filament buckles depends on the rigidity of the filament, the amount of force generated by the motors, and the distance $a$ between the connectors. Under conditions in which the filaments always buckle, there are no expansile configurations, and the net sum is $-P_M P_C (1-P_C) v$. In this simple

case, the sign reveals that the system is contractile. Moreover, the predicted contractile rate is nonzero if both $P_M$ and $P_C$ are nonzero, which is the case when both motors and crosslinkers are present. The formula also shows that when crosslinkers are added (*i.e.*, as $P_C$ changes from 0 to 1), the contractility increases and eventually vanishes. Contractility is thus maximal at an intermediate quantity of crosslinkers.

### Contraction rates of networks of semi-flexible filaments

If the filaments are semi-flexible, which is the case for F-actin, with a rigidity of 0.075 pN μm², the contribution of expansile configurations may not always be negligible, since a filament may or may not buckle depending on the length over which it is compressed. Therefore, to be able to predict the behavior of a network, it is necessary to know the conditions under which filaments buckle.

For an empirical assessment of this effect, we thus simulated networks in which the length and density of the filaments, and the number of crosslinkers and molecular motors were systematically varied. For this, we used Cytosim, an Open Source simulation engine that is based on Brownian dynamics (Nedelec & Foethke, 2007). In brief, each filament is represented by a set of equidistant points, subject to bending elasticity (Box 2A). Crosslinkers and motors are represented by diffusing pointlike particles, which bind stochastically to neighboring filaments (Box 2C and D). Connectors with a stiffness $k$ are formed when motors or crosslinkers are bound to filaments with each of their two subunits (Box 2E). The movement of motors follows a linear force–speed relationship (Box 2F). For simplicity, the unbinding rate is constant for this study, and a motor reaching the end of a filament immediately unbinds

(Box 2G). Given a random network as initial condition, Cytosim simulates the movement of all the filaments in the system (Fig 2B, Movies EV1–EV3), and a contraction rate is extracted automatically (Appendix Supplementary Methods C).

Guided by the results of many simulations, we concluded that network contraction depends on the threshold distance $b$ above which buckling occurs, which in turn can be calculated from the filament rigidity and the maximum force exerted by the motor. If $L_1 < b$, then any filament segment of length $b$ will be intersected by $\beta_0 = b/L_1$ filaments, where $L_1$ is the mesh size of the network. If any of these intersections is bridged by a crosslinker, this fixes the filament laterally and prevents it from buckling under the force of the motor(s) and crosslinker positioned at its ends (see Appendix Supplementary Methods G for more details). From these considerations, we can calculate the probability for a filament segment to buckle as $P_M P_C (1-P_C)^{\beta_0}$, where $(1-P_C)^{\beta_0}$ is the probability that all the intersections between the motor and the crosslinker are free of crosslinkers. With this adjustment, the theory correctly predicted the dependence of the contraction rate on the number of connectors for a variety of conditions (Fig 2C and D, Movie EV4). It also predicts previous results where contraction of *in vitro* actomyosin networks was obtained only in the presence of both crosslinkers and motors (compare Fig 2D with Fig 2D from Bendix *et al*, 2008).

### Contraction and expansion of networks of rigid filaments

We next explored systems composed of rigid filaments such as microtubules. Because some molecular motors are associated with microtubule ends in nature, we investigated the behavior induced

---

**Box 2: Elements of the stochastic model of cytoskeletal dynamics.**

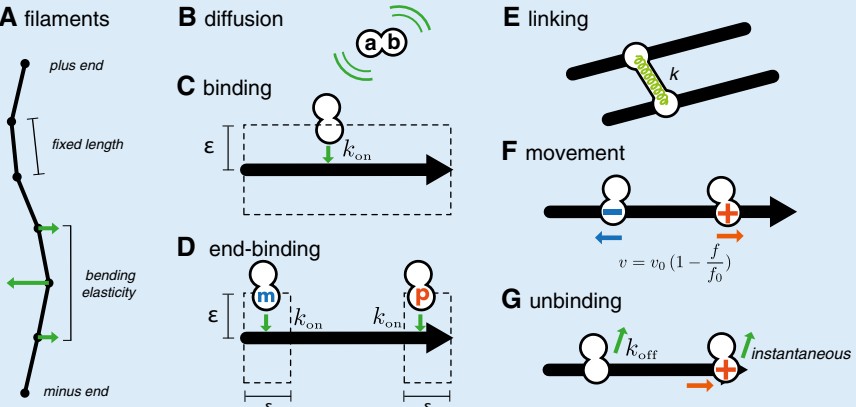

(A) Networks of flexible filaments are simulated using a Brownian dynamics method. Filaments are polar, thin and have a constant length. Each filament is modeled with an oriented string of points, defining segments of equal lengths. The movement of filament points follows Brownian dynamics, with elastic forces such as the bending elasticity of the filament, and the elasticity of connectors. (B) In the simulations, connecting molecules are made of two independent filament-binding subunits (a and b, which can be any one of those defined in Fig 1B). When both subunits are unattached to filaments, the molecule diffuses within the simulation space. (C)

Binding occurs at a constant rate $k_{on}$ to any filament closer than ε. Attachment occurs on the closest point of the filament. (D) End-binding follows the same rules as binding, but is restricted to a distance δ from the targeted filament end. (E) Connectors act mechanically as Hookean springs between two filaments, with stiffness k and zero resting length. (F) Motor subunits move toward either the plus- or minus-end of the filament with a linear force–velocity relationship. (G) All connector subunits detach with a force-independent rate $k_{off}$, and motors detach immediately upon reaching a filament end.

---

 

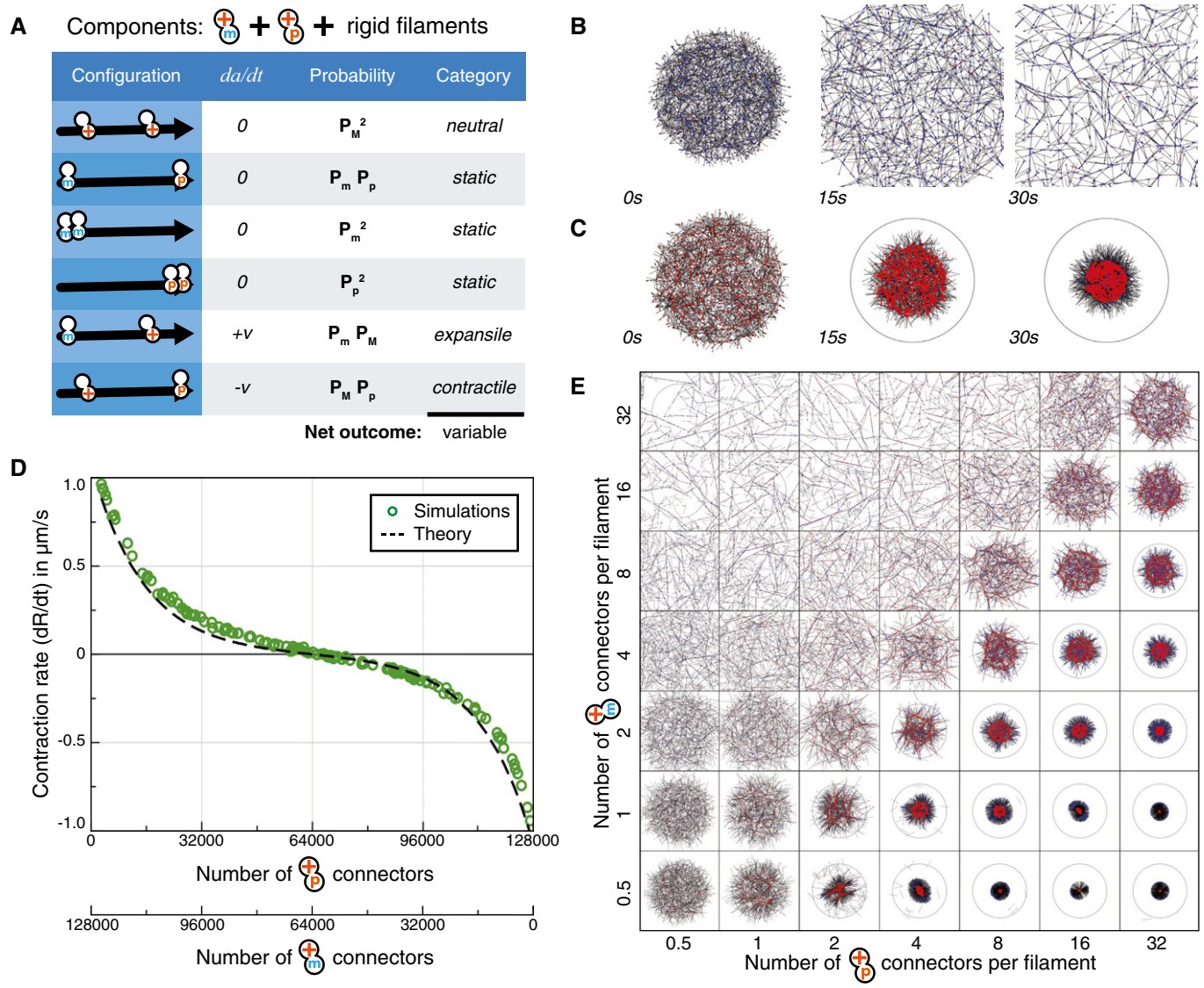

**Figure 3. Predictions and simulations for microtubule-like networks of rigid filaments.**

A   A system composed of rigid filaments and two types of connectors. One connector consists of a plus-end-directed motor combined with a minus-end binder, the other is a plus-end-directed motor combined with a plus-end binder. There are six possible configurations involving these two connectors.

B   Three time points during the evolution of an expansile network of 1,500 straight filaments (their bending rigidity is set as "infinite" here) with 1,500 motor/plus-end binders and 48,000 motor/minus-end binders initially distributed over a circular area of radius 15 μm.

C   Three time points during the evolution of a network similar as (B), but with 48,000 motor/plus-end binders and 1,500 motor/minus-end binders.

D   The contraction rate of a network as a function of the numbers of the two types of connectors, which are inversely varied. Each symbol represents a simulated random network of 4,000 straight filaments initially distributed over a circular area of radius 25 μm. Details of methods as in Fig 2C. The broken line indicates the analytical prediction made by the theory (Appendix Supplementary Methods G).

E   Simulations of networks containing varying numbers of connectors. Networks contain 1,500 filaments initially distributed over a radius of 15 μm. Depending on the concentrations of the connectors, the network can be expansile (top left corner) or contractile (bottom right corner). Snapshots at *t* = 30 s.

Source data are available online for this figure.

by connectors comprised of motors and end-binding subunits (Fig 3). As predicted by the theory (Fig 3A), the simulations showed that the system is expansile if plus-end-directed motors are combined with minus-end-binding subunits (Fig 3B, Movie EV5), and contractile if plus-end-directed motors are associated with plus-end-binding subunits (Fig 3C, Movie EV6). A system composed of these two types of connectors can be either contractile or expansile

depending on the relative concentrations of the connectors (Fig 3D and E, Movie EV7).

### Prediction of the effects of combinations of connectors

To probe the general applicability of the theory, we simulated networks with mixtures of connectors containing five different types

of subunits (Fig 1B). A subunit can bind, and then either remain bound at the initial position, or move. Non-moving binders may be of a type that can bind anywhere on the filament, or they may be restricted in their binding to a region near the plus or the minus end. Moving elements (motors) can bind anywhere, but can be of two types, those moving to the plus and those moving to the minus end. By combining any two of these subunits, one can make 15 types of connectors. Simulated networks containing any one type of connector all behaved as predicted by the theory (see examples in Fig 4A). We also simulated systems containing two different types of connectors (in equal quantities), both for flexible and rigid filaments. There are 210 possible combinations, and for every one of them, the simulations closely matched the behavior predicted by the theory (Fig 4B and C, see Appendix Supplementary Methods F for details of the calculation). Many types of molecular elements that are found in nature, such as end-binding proteins, have not been used in reconstituted networks, but we can now predict what their effects on a network should be.

**Heterogeneous systems composed of different types of filaments**

So far, we have considered systems made of one type of filament, but some networks *in vivo* contain different types of filaments. A prominent example are the thick antiparallel "minifilaments" with a length of 300 nm formed by myosin II motors (Verkhovsky & Borisy, 1993). Networks such as the actomyosin meshwork of the cell cortex and the contractile actin cables in cells are thus heterogeneous systems in which F-actin filaments are mixed with minifilaments, which also harbor the motors driving the system out of equilibrium. To probe if the theory could hold for such heterogeneous systems, we listed all the possible combinations of two connectors for the two types of filaments (Fig 5A and B). Similar to the homogeneous case (Fig 2), this analysis predicts that the system should be contractile if crosslinkers are also present, and neutral otherwise. We then simulated such a system of actinlike filaments and minifilaments composed of a rigid backbone of length 0.5 μm with a motor subunit at each end. The results confirmed the predicted behaviors (Fig 5C and D), suggesting that the theory can be applied to heterogeneous networks.

**Effect of filament turnover on contractile systems**

So far, we have considered systems made of filaments of fixed length that persist indefinitely. Under these conditions, network contraction and expansion are non-reversible events, and they occur only once. This is indeed what happens with most *in vitro* reconstituted cytoskeletal networks obtained with stabilized filaments (Takiguchi, 1991; Katoh *et al*, 1998; Surrey *et al*, 2001; Thoresen *et al*, 2011; Murrell & Gardel, 2012; Carvalho *et al*, 2013; Vogel *et al*, 2013; Shah *et al*, 2014; Foster *et al*, 2015). But how does this relate to networks *in vivo*, which manage to avoid such a collapse? The simulations described above do not correspond perfectly to the situation *in vivo*, because cytoskeletal filaments are dynamic, such that both the length of the filaments, and their abundance are fluctuating quantities that can be regulated. Contractile cortical networks often do not simply contract monotonically and irreversibly, but can show dynamic contractile foci, with pulsatile contractions persisting over extended periods (Munro *et al*, 2004; Martin *et al*, 2009; Solon

*et al*, 2009; He *et al*, 2010). To test the relationship between this dynamic behavior and contractility, we extended our simulations to include filament turnover. Instead of making the filaments shrink and grow, we modeled turnover by simply taking out individual, random filaments *in toto* and replacing them by new ones. To implement an average lifetime T for the filament, we randomly selected and deleted one of the N filaments at a rate N/T, and replaced it with a new one placed at a random position (Fig 5E). However, circular networks of the type we have considered so far still contracted into a central focus even with filament turnover. We thus implemented a model with periodic boundary conditions (PBC). Use of PBC imposes a constant surface on the system and thereby forces the network to build up tension. It corresponds best to a network that is attached at the cell boundaries, without requiring additional assumptions on the nature of the attachment. Under these conditions, filament turnover had a significant effect on the contractile behavior. We observed that for 3 s < T < 200 s, most configurations that had been contractile without turnover now displayed pulsed contractions (Fig 5F, Movie EV8). These results confirm earlier models that considered filament dynamics (Bidone *et al*, 2017) or turnover (Hiraiwa & Salbreux, 2016), illustrating that with filament turnover, a system that was contractile otherwise can be pulsatile. As suggested by an active gel theory (Kumar *et al*, 2014), we wondered whether pulsatility was a general consequence of turnover. We thus simulated networks that were contractile on Fig 4B and varied systematically the filament turnover rate. Most displayed pulsatile behavior for a certain range of values (See Dataset EV1). Thus, pulsatility appears to be a common consequence of filament turnover, irrespective of the type of connectors in the network.

# Discussion

The theory we present here predicts the initial evolution of a network from the properties of its connectors. We have confirmed these predictions with simulations for all tested conditions. The model implemented in the simulation is intentionally minimalistic, with subunit binding, unbinding and filament turnover occurring at constant rates and independently of other events. All simulations were done in 2D and did not consider steric interactions between the filaments, which in 2D would induce artifacts. We expect our theoretical arguments to hold also for other types of networks such as filament bundles or 3D networks. However, the calculation presented in the Appendix depends on the geometry of the network and would need to be revised to apply to different geometries. It is tempting to think that the approach can also be extended to anisotropic networks if the probabilities of the configurations are calculated locally.

For simplicity, we simulated homogeneous networks with a circular geometry. While this may not represent the usual *in vivo* situation, it corresponds precisely to networks made recently *in vitro* using light deactivation of the myosin inhibitor blebbistatin (Linsmeier *et al*, 2016; Schuppler *et al*, 2016). Our analytical prediction of network behavior was based on the characteristics of the connectors but did not include the viscosity of the medium. This is because we implicitly assumed that the motors were moving at constant speed, or in other words that they were operating far from

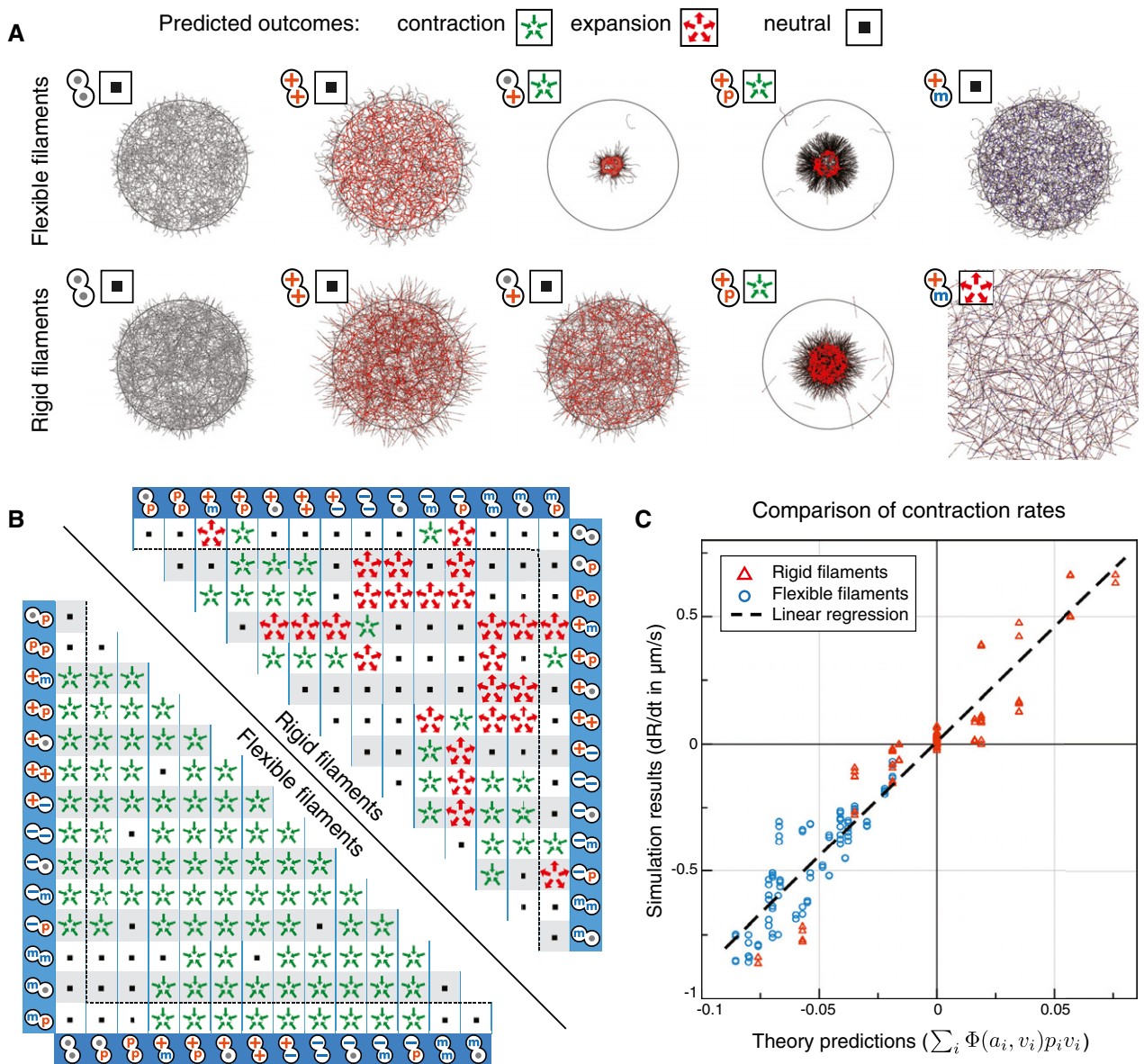

**Figure 4. Additional predictions of the theory.**

The predictions of the theory for the various conditions shown here are represented graphically as sets of green centripetal arrows for contraction, red centrifugal arrows for expansion, and gray squares for neutral networks.

A Examples of simulations of networks with the indicated types of connectors. The predicted outcomes of network contraction, expansion, or neutrality (symbol at the top left of each simulation) are confirmed in each case by the behavior of the network in simulations. The networks are composed of 1,500 flexible or rigid filaments, and 24,000 connectors. Snapshots at *t* = 20 s.

B Summary of the predictions for random networks with all possible combinations with two types of connectors, either with flexible (bending rigidity = 0.01 pN μm², below diagonal) or rigid filaments (infinite rigidity, above diagonal). The networks contain 4,000 filaments and 64,000 connectors, 32,000 of each connector type, indicated by the labels of the corresponding row and column. These results were generated using Preconfig (Nedelec, 2017).

C Comparison of the contraction rates predicted by the theory (horizontal axis) with the rates obtained by simulation (vertical axis). Each data point indicates one of the 210 systems considered in (B). Networks are made of 4,000 filaments and 64,000 connectors initially distributed over a circular area of radius 25 μm. In this case, all the binding parameters of the subunits and the concentration of connectors are always equal, such that the prediction is simplified (Appendix Supplementary Methods E).

Source data are available online for this figure.

their stall force and that filament drag forces were insignificant. The procedure depicted in Box 1 provides the absolute contraction rate of the network. The formula (Box 1B) is useful even if not all of the microscopic quantities are known. For example, if the length of the filaments is unknown, one can still calculate the numerator of the fraction defining $\chi$ (Box 1B) to predict how contraction rates are affected by changes in the connectors (numbers, types, binding rates, unbinding rates), as done for Figs 2C and 3D. Such a

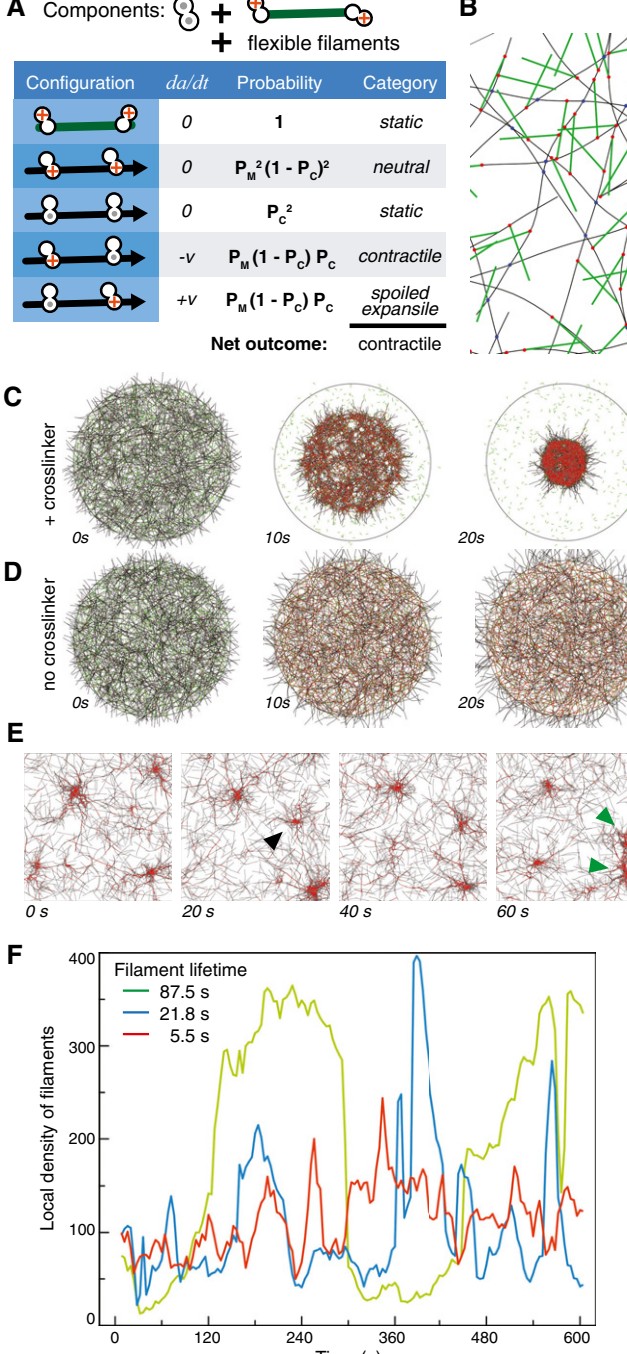

**A** Components:

**A** flexible filaments

| Configuration | $da/dt$ | Probability | Category |
|---|---|---|---|
| | 0 | 1 | static |
| | 0 | $P_M^2 (1 - P_C)^2$ | neutral |
| | 0 | $P_C^2$ | static |
| | $-v$ | $P_M (1 - P_C) P_C$ | contractile |
| | $+v$ | $P_M (1 - P_C) P_C$ | spoiled expansile |
| | | **Net outcome:** | contractile |

**B**

**C** + crosslinker

*0s*      *10s*      *20s*

**D** no crosslinker

*0s*      *10s*      *20s*

**E**

*0 s*      *20 s*      *40 s*      *60 s*

**F**

Filament lifetime
— 87.5 s
— 21.8 s
— 5.5 s

Local density of filaments / Time (s)

**Figure 5. Heterogeneous and pulsatile systems.**

A   Configurations present in a heterogeneous network containing rigid minifilaments and flexible actin-like filaments. The motors are permanently attached at the extremities of the minifilaments, so as to represent myosin minifilaments. The system is predicted to be contractile in the presence of passive crosslinkers connecting actin filaments directly, and neutral without crosslinkers.

B   Detail of a simulation with minifilaments (green) and crosslinkers (blue).

C, D   The simulated systems contract only if crosslinkers are included.

E   Time series of a simulation with filament turnover, 1,400 filaments (rigidity 0.075 pN µm²), 22,400 motors, 5,600 crosslinkers within periodic boundary conditions with size 16 µm. Filament turnover was implemented by deleting a randomly selected filament and placing a new filament at a random location, stochastically with a rate $R = 64$ s⁻¹, corresponding to an average filament lifetime of ~21.8 s. The series shows the formation of a new contractile spot (black arrowhead) and its downward movement and fusion with another contractile spot (green arrowheads).

F   The local density of filaments in an arbitrarily chosen region covering ~6% of the simulated space as a function of time. The data with filament lifetime 21.8 s are from the simulation shown in (A). The network continues to redistribute, showing irregular variations of the local filament density, and does not collapse into one spot.

Source data are available online for this figure.

prediction is immediately valuable, as it can be readily tested experimentally by systematically varying the concentrations of both motors and crosslinkers in reconstituted *in vitro* networks. As a network evolves distinct patterns such as asters, bundles or vortices can arise which can significantly change the network dynamics. Our theory does not deal with the formation of such patterns, but remains valid as long as the evolving network is not strongly anisotropic and remains percolated.

For a system containing crosslinkers and bifunctional motors, our analysis indicates that the "active" contractile configurations must contain both a crosslinker and a motor. We thus expect these two types of element always to be found in a contractile system. Contractile systems have been reconstituted *in vitro* to study this point, but it is important to remember that an assumedly pure preparation of motors that is added may in fact contain damaged "dead" motor proteins that act as passive connectors. Thus, a mixture that is assumed to contain only filaments and motor proteins may in fact also contain some crosslinkers. Even so, addition of crosslinkers indeed dramatically enhances the effect of myosin, a phenomenon observed more than 50 years ago (Ebashi & Ebashi, 1964). For 2D networks, the theory explains why the maximum contractility is obtained *in vitro* with approximately equal amounts of motors and crosslinkers (Bendix *et al*, 2008; Köhler & Bausch, 2012; Ennomani *et al*, 2016), and why, in the absence of crosslinkers, networks fail to contract despite the presence of molecular motors, as reported by Bendix *et al*, 2008. Our theory also explains that under the action of myosin VI, a branched network made with Arp2/3, which represents an example of a connector consisting of an end-binding and a side-binding component, is more contractile than a network connected by crosslinkers that bind anywhere along the filaments (Ennomani *et al*, 2016). Because myosin VI is directed to the minus (pointed) end, the configuration containing a crosslinker bound to the minus end (Arp2/3) is always contractile. Thus, at equal levels of connectivity, a network made with Arp2/3 and myosin VI is more contractile than a network made with a non-specific crosslinker, and less contractile than a network made with only end-to-end crosslinkers. Conversely, we predict that the effect would be opposite in the presence of plus-end-directed motors, like myosin II: An Arp2/3 network should be less contractile than a network of equal connectivity made with non-specific crosslinkers.

Our finding that introducing filament turnover was sufficient to induce pulsing in most of the contractile scenarios leads to the surprising conclusion that pulsatility may be an intrinsic behavior of contractile networks made of non-stable filaments and that no other external triggers are necessary. This of course does not mean

that in the natural biological situation there may not be regulatory elements superimposed on the underlying mechanism that suppress or enhance pulsing (Nishikawa *et al*, 2017). Pulsing is seen only over a certain range of filament lifetimes, indicating that one such regulatory input could be via the stability of filaments: For example, increasing the stability of filaments should, according to our simulations, arrest or reduce pulsing, whereas the ability of myosin to destroy filaments (Matsui *et al*, 2011) could lead to enhanced pulsing. Many parameters of the actomyosin network can tune the characteristics of the pulses, as has been shown for myosin (Munjal *et al*, 2015). While pulses appear to be an inevitable consequence of filament turnover, their importance for the biological functions of cytoskeletal network needs to be clarified.

The theory presented here unifies previously proposed mechanism for a number of biological systems, and we will discuss now how various contracting or expanding systems can be represented and their behavior predicted within the new theory (Box 3).

In the sarcomeres found in striated muscles, myosin II minifilaments pull on filaments arranged in an antiparallel manner (Box 3A). This system can be seen as containing two types of connectors: a passive one linking the barbed (plus) ends of the filament and a motor directed toward the barbed (plus) end. Three possible combinations can be made with these two connectors (Box 3, right column). Because none of these configurations is expansile, the system is bound always to be contractile. Even if they are not as highly ordered as a sarcomeric system, less organized systems made of the same subunits, for example, bipolar filaments in smooth muscles (Box 3B) are also contractile.

For a system in which the crosslinkers are not restricted to binding to the filament ends, but can bind anywhere along the length (Box 3C) both contractile and expansile configurations arise. Following the discussion on how buckling promotes contraction of a disorganized actin network (Mizuno *et al*, 2007; Liverpool *et al*, 2009; Lenz *et al*, 2012), we argued that buckling can spoil some of the expansile configurations, tipping the balance in favor of contraction.

One mechanism to explain the contraction of microtubule networks (Box 3D) does not require filament bending, but involves a motor that can halt at the end of the filaments (Hyman & Karsenti, 1996; Nedelec *et al*, 1997). Because the motor walks toward the end, where it may be transiently trapped, configurations are contractile or neutral, but never expansile, and the entire network itself is therefore contractile (Foster *et al*, 2015). Looking at the set of configurations (Box 3, right column), the similarity of this mechanism with sarcomeric contractility (Box 3A and B) becomes apparent. In the case of the end-dwelling motor, however, the same molecular type is involved in generating the active and neutral end-binding connections.

Although we did not consider filament disassembly in this study, the theory can be applied also to this situation. For example, a molecule that tracks and remains bound to the depolymerizing end of a filament (Box 3E) will reduce the distance between itself and a connector located elsewhere on the filament, thereby creating a pulling force. By calculating the likelihood of such a configuration, one may be able to predict the overall contractility of the network. We also did not consider filament elongation, which is a prominent mechanism by which actin networks expand.

A system with expansile configurations can only extend if the filaments are sufficiently rigid to resist buckling, which depends on the density of the network, and is more likely to be the case for microtubules than for actin. We will discuss two examples of expansile microtubule systems: the mitotic spindle and the marginal band of blood platelets. A mitotic spindle evolves throughout the cell division cycle, but during metaphase, it usually keeps a constant length. To maintain this steady state, contractile and expansile forces must be kept in equilibrium. In *Xenopus laevis*, contraction is driven by dynein (Foster *et al*, 2015) and other minus-end-directed motors, while expansion is driven by the plus-end-directed motor kinesin-5 (Needleman & Brugués, 2014). When anaphase is induced, the metaphase balance is broken and the spindle elongates. Can this be explained by the theory? For the sake of the argument, let us assume that the function of dynein ceases completely, and ignore minus-end-directed motors altogether. We then need to consider only two types of connectors (Box 3F): passive complexes containing the protein NuMA, which connect the minus ends of microtubules at the spindle poles, and plus-end-directed motors kinesin-5 connecting adjacent, antiparallel microtubules. Since kinesin-5 moves away from the minus ends, the model indeed predicts that the anaphase spindle is expansile, using configurations that are symmetric to the sarcomeric systems (Box 3A). A disorganized network made of the same connectors would also be expansile.

Other expansile microtubule systems can be found in blood platelets and their progenitor cells, the megakaryocytes. During proplatelet generation, the microtubules assemble into bundles that elongate under the activity of the molecular motors dynein (Patel, 2005). In the mature platelets, microtubules are organized into a closed circular bundle which must be able to resist contractile forces as it pushes outward on the plasma membrane. It was recently reported that the microtubule ring elongates after platelet activation, in a manner that is dependent on microtubule motors, but the mechanism that drives elongation is still unclear (Diagouraga *et al*, 2014). Our systematic exploration of random networks (Fig 4) suggests different scenarios that could explain why this system is expansile. Beyond the relevance to these *in vivo* systems, it will be exciting to follow these principles to create expansile networks of microtubules *in vitro*, since end binders are available to synthetic biologists. Figures 3 and 4 suggest exciting avenues for the development of synthetic materials (Henkin *et al*, 2014) that could be tuned to be expansile or contractile, which could be achieved, for example, by using light-switchable molecular motors (Nakamura *et al*, 2014).

Finally, in a system where the symmetry provides an equal number of contractile and expansile configurations, any imbalance in the probabilities of these configurations may lead to overall contraction or expansion (Gao *et al*, 2015). Following this principle, we can suggest here an explanation for the expansile nature of *in vitro* microtubule networks (Sanchez *et al*, 2012). Particularly, if the motors are sufficiently processive, they may run a distance that is comparable to the length of the filament, and in this case, their distribution along the length of the filament will be non-uniform (Box 3G). This effect has been called the antenna effect (Varga *et al*, 2006), and arises as a consequence of the motility, in a situation where binding has the same probability at every position of the filament. A plus-end-directed motor would become enriched near the plus ends of microtubules (Box 3G). In the presence of crosslinkers (that can be dead motors), such an effect will increase the likelihood of the expansile configurations, and lower the likelihood of the contractile configurations, thus promoting expansion. Even if

**Box 3:    Review of contractile and expansile mechanisms.**

Previously described mechanisms can be represented in the terms of the theory by focusing on pairs of connectors present on filaments. The sarcomeric mechanism (A), and an analogous mechanism involving bipolar filaments (B) each have a plus (barbed)-end-directed multivalent motor acting on filaments that are connected at their minus (pointed) ends by molecular complexes that act as connectors. These systems are always contractile because there are only two active configurations: one involving two motors, which is neutral, and one with a motor and an end binder, which is contractile since the motor always moves toward the end binder. The buckling-dependent mechanism (C) leads to contractility because the flexibility of the filament spoils the expansile configuration. Thus, if the filaments are sufficiently flexible, the net effect will be contractile (see Fig 2). In systems containing only end-dwelling multivalent motors (D), the motors generate contraction without added passive connectors, because they eventually come to a halt at the end of the filament and thereby act as end-binding connectors. Configurations involving a motor halted at the end, and a motor moving toward this end along the same filament result in contraction. There is no expansile configuration in this mechanism, and the net effect is therefore always

contractile, irrespective of filament buckling. A connector with a subunit that binds to a disassembling end of a filament (E) generates only one active configuration, which is always contractile, even in the absence of motors. In this example, the end-tracker binds to the plus end and moves toward the minus end by tracking a depolymerizing end (or inducing its disassembly). (F) A mitotic spindle at anaphase may be considered as a network held together by multivalent plus-end-directed motors from the kinesin-5 family, and by factors connecting the microtubule minus ends at the spindle pole. With these two types of connectors, the configurations involving two connectors are neutral, static, or expansile. (G) A system can be made expansile by the "antenna effect", because motors acquire an asymmetric distribution profile along the filaments. In the presence of this effect, contractile configurations are less likely than expansile ones, and the overall system can become expansile as a consequence. (H) Some mechanisms of contraction involve two connectors acting on more than one filament. In the case depicted here, two crossing filaments will be "zipperred together", by a pair of connectors moving apart. This configuration is able to create a contractile force dipole in a direction perpendicular to the filaments.

the motors were directed toward the minus ends of microtubules, the antenna effect would still lead to a bias in favor of expansion. The net imbalance will depend on the biophysical properties of the motors (speed, unbinding rate), and the length of the microtubules, and could provide tunable expansibility for networks (Sanchez *et al*, 2012).

In conclusion, our theory offers a framework for elementary mechanisms of expansion or contraction. It is a starting point for further exploration, since in its current state, the theory does not explain all the phenomena observed in simulations. For example, if two flexible filaments are connected by two connectors with one moving away from the other, this can contribute to contraction (Box 3H). The connectors can pull the ends of the filaments, and therefore the network together, even though the distance between them is growing. This interesting effect, which is analogous to a zipper, can only be understood by considering two filaments and two connectors, whereas our theory considered one filament and two connectors. In the networks studied here, this mechanism has only small effects (see Appendix Supplemental Materials G).

From the theoretical framework presented here, with its clear predictions, perhaps a classification of the different types of active networks found in nature will emerge. Our approach may also inspire novel avenues for synthetic filament networks with enhanced functionalities.

**Expanded View** for this article is available online.

## Acknowledgements
We are thankful to EMBO and EMBL for support, in particular for its high-performance computing services. We thank members of the Leptin and Nedelec groups, particularly H. Turlier, S. Dmitrieff, M. Lera Ramirez, and Y. Jeske, as well as S. Blandin, D. Gilmour, T. Hiiragi, J.-P. Shen, R. Prevedel, and P. Lenart for critically reading the manuscript. J.M.B. is the recipient of an EMBL Interdisciplinary Postdoctoral Fellowship, which is co-funded by Marie Curie Actions of the European Commission.

## Author contributions
This work arose from regular discussions between the authors, which all have contributed significantly to the findings. All authors together wrote the article.

## Conflict of interest
The authors declare that they have no conflict of interest.

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
