## [Review Process File · Molecular Systems Biology]

A theory that predicts behaviors of disordered cytoskeletal networks

Julio Belmonte, Maria Leptin & François Nédélec

Corresponding author: François Nédélec, European Molecular Biology Laboratory

Review timeline:

Submission date:	06 June 2017
Editorial Decision:	04 July 2017
Revision received:	24 August 2017
Editorial Decision:	31 August 2017
Revision received:	31 August 2017
Accepted:	05 September 2017

Editor: Thomas Lemberger

Transaction Report:

1st Editorial Decision

04 July 2017

Thank you again for submitting your work to Molecular Systems Biology. We have now heard back from the three referees who accepted to evaluate the study. As you will see, the referees find the topic of your study interesting and I am pleased to inform you that they are all supportive.

The reviewers raise a few points that we would invite you to consider in a minor revision of this work:

- More details on some of the analytical derivations would be welcome (see reviewer #2 point#2)
- Reviewer #2 had difficulties in understanding why the theory predicts only relative contraction rates. This should be clarified.
- Some of the assumptions may still need to be more explicit (see for example referee #3, point #2, rev #1 pt #1)
- Finally, if possible, the origin of pulsatile contractions might need further explanations (ref #1, point #3)

For the simulations shown in the figures or as movies in this paper, we would invite you to include the Cytosim (.cym) files with the respective parameters used to run the depicted simulations. These files can be zipped (ideally with a README file) and uploaded as datasets labeled "Computer simulation file (Cytosim) for Figure XX-ZZ" ..

REVIEWER REPORTS

Reviewer #1:

This is an exciting theoretical study that examines contractility of actin-myosin networks. In fact, the authors consider very general class of polymers contracted by ensembles of various types of motors and crosslinkers. The big question, which pre-occupies the field lately, is whether and how myosin can contract random actin arrays - where does an asymmetry, necessary because myosin contracts antiparallel filament pair overlapping near their minus ends, but expands the pair overlapping near their plus ends, come from. A number of theories was proposed - some (Lenz/Gardel et al) posit that there is a mechanical asymmetry because expanding pair buckles, other (Oeltz/Mogilner et al) - that the asymmetry is due to polar filament treadmilling, yet other (Lenz/Carlsson et al) - that zippering or rotation in 2D is responsible for this asymmetry. What plagues all these theories is that analytical estimates they produce are not very reliable, while numerical simulations provide limited insight.

The powerful approach of the present study is to consider all possible intersections of a filament with other filaments, all combinations of motors and crosslinks at these intersections, and then use elementary mechanics and probability estimates to derive analytically rules for whether there is a contraction or expansion or oscillations of the networks depending on the conditions.

The analytical results are in a shockingly good agreement with numerical simulations. What this means is that there is a comprehensive phase diagram that for the first time allows us to predict the contractile behavior based on densities, filament flexibilities and length distributions.

This is a great result, broadly applicable and interesting, very systems-bio-like, and appropriate for MSB. The manuscript is written very well.

The comments and questions below are not a criticism, but rather something for the authors to consider:

- 1) as contraction goes on, many theories predict that with time patterns will form in the network that would worsen the contraction. Presumably, numerical simulations then should be fitting the analytical predictions worse? can this be tested?
- 2) For very dense networks, theory probably misses multiple intersections per filament. What does numerical simulation predicts in this case - worse contraction?
- 3) There has to be a better qualitative explanation for pulsatile contraction - looking at the movie and figure I see that the effect is due to transient local filament clusters, but what are the essential feedbacks that always underlie oscillations, and what explains the characteristic time scales.
- 4) there is some, but very sketchy 2D vs 3D network discussion - would be good to see some semi-quantitative estimates of expected differences.
- 5) Finally, theories of Lenz, Carlsson, Mogilner, others, have some quantitative predictions - how do those compare to the theory in this study?

Reviewer #2:

This manuscript presents a framework for understanding if mixtures of motors, cross-linkers, and filaments will be contractile or expansile. This is a significant advance that helps unify a wide variety of previous proposals. I believe that this is very important work that provides great insight into the collective properties of the cytoskeleton.

I have a number of suggestions that may help to improve the manuscript.

MAJOR POINTS:

1) The central equation of the presented theory, Box 1, parts A and B, contains the variable v_i , which is the rate of change of the spacing between two motors/crosslinkers. Given that motors have a force velocity relationship, it is unclear how v_i is determined in analytical calculations. Is v_i given from the unloaded motor velocity v_m ? If so, does that imply that the motors are under low load, which in turn suggest that this theory may fail for large or heavily crosslinked networks. If not, it is unclear how v_i is determined.

I think that this issue may be related to, what was to me, a cryptic statement in the Discussion:

"Rather than an absolute contraction rate, this theory predicts the relative contraction rates of two networks when the parameters of the connectors (numbers, types, binding rates, unbinding rates) are different between them." It is unclear to me why the author's theory cannot be used to predict absolute contraction rates, or given that, why it does successfully predicts relative contraction rates. It is very important to explain this issue in more detail.

2) In the author's discussion of networks of semi-flexible polymers, they present the results of an analytical calculation (Figure 2C), without giving the detailed calculation that was used to obtain the plotted line. They says that this aspect of their theory was "Guided by the results of many simulations" without presenting the results of those simulations. The authors really do need to provide more details on this aspect of their theory. They should give a full derivation of their analytical calculations and present the simulations that support the validity of their calculations. For example, they should present simulations in which filament density and persistence length are systematically varied and show the extent to which their analytical theory can or cannot these results.

MINOR POINTS:

1) At some times the authors focus on the effects of the crosslinkers on the network (as very well illustrated in the first three part of Figure 1E), while at other times the authors focus on the effects of the crosslinkers on the filament (as in the "effect" label on Figures 2A and 3A). This distinction is not clearly explained, which makes the presentation confusing at points.

2) The paper would read better if much of the speculation and possible future directions were moved to the discussion section. For example the points about light-switchable motors on line 197, and the line about anisotropic networks on line 232 seem out of place.

3) The authors state that "We thus simulated networks with all the combinations of connectors that were contractile on Fig. 4B and varied systematically the filament turnover rate. All displayed pulsatile behavior, confirming the universal nature of the phenomenon (data not shown)." It is difficult for a reader to evaluate claims of universality without more details. The authors need to present these results. How did they vary filament turnover? Over what range of values? How exactly did they determine that these networks were pulsatile?

4) Figure 4B is difficult to read, especially in black and white, perhaps a different color set (besides red and green) and a different set of symbols would work better. The difference between inward point arrows and outward pointing arrows is also difficult to notice when the paper is printed.

5) On line 182, the use of the word "bivalent" is confusing. Is it intended to contrast with monovalent or trivalent motors?

Reviewer #3:

In the manuscript entitled "A theory that predicts behavior of disordered cytoskeletal networks" the authors use a combination of theoretical modeling and computer simulations to investigate the dynamical behavior of biological networks comprised of active molecular motors, passive cross-linkers and elongated filaments. Previously, the properties of active networks have been extensively studied using coarse-grained hydrodynamic models. However, such models cannot connect microscopic dynamics of constituent unit to the macroscale behavior. Consequently, this is an important subject area that remains poorly understood. The authors describe a comprehensive study. They propose a simplified theoretical model that predicts how the contraction/extension of an active network will depend on the properties of the constituent units. Computer simulations quantitatively test the theoretical predictions over an impressively wide range of molecular parameters. I believe that the manuscript is of high quality and acceptable for publication. I find it quite surprising how a very simple model describes network dynamics over a wide range of parameters. Overall the manuscript is well written but addressing the following few queries would further clarify and

improve it.

1. Can the proposed theoretical model quantitatively describe the rheological properties of passive networks? This might be essential for comparison to experimental data. For example, experiments on the contracting actin-myosin gels (Bendix et. al. Biophysical Journal 2008) show that myosin molecular motors can induce macroscopic contraction only at intermediate cross-linker concentrations. Presumably at high cross-linker concentrations the stiffness of the network is too high for motors to induce network contractility. It is not obvious that the model captures this effect. This is not essential for publication but it would be useful to comment on.

2. The authors state: "to predict if the whole network will contract or expand, we sum up the effect of all elementary mechanical bridges in the network." I think that the idea that the macroscopic contraction or expansion can be directly related to dynamics of single filaments is intriguing. However, I would like to see more detailed explanation of how one relates contraction of individual segment to the overall network contraction. Furthermore, their model involves a number of assumptions that should clearly be stated in the text. For example, I believe that the model assumes that the network structure remains the same throughout the entire contraction process. This is not necessarily the case in experiments, as the filaments can form bundles at high crosslinker concentrations or orientationally ordered nematics. How does the model account for such possible dynamical pathways?

3. The senior author of the submitted manuscript has previously shown in an important advance that clusters of molecular motors and filaments can assemble into mesoscopic structures such as asters and vortices. Can the current model predict formation of such structures? This should be discussed.

4. The model assumes that there are no excluded volume interactions between filaments. In absence of such interactions, what ultimately determines the final density of the contracting network?

Dear Thomas,

We are pleased to know that all the referees found the work interesting and we thank them for their support. We have followed their suggestions to improve the manuscript, and we hope that the revised manuscript is suitable for publication. Below you will find our answers to each referee comment in blue. We have also prepared additional online material, in particular the Cytosim configurations files, that will allow anyone to repeat the simulations presented in the article with our Open Source software.

Sincerely yours,
Francois Nedelec

Dear François,

Thank you again for submitting your work to Molecular Systems Biology. We have now heard back from the three referees who accepted to evaluate the study. As you will see, the referees find the topic of your study interesting and I am pleased to inform you that they are all supportive.

The reviewers raise a few points that we would invite you to consider in a minor revision of this work:

More details on some of the analytical derivations would be welcome (see reviewer #2 point#2)

We have added a new section G to the Appendix material to describe the analytical calculations in more detail.

Reviewer #2 had difficulties in understanding why the theory predicts only relative contraction rates. This should be clarified.

- Some of the assumptions may still need to be more explicit (see for example referee #3, point #2, rev #1 pt #1)
- Finally, if possible, the origin of pulsatile contractions might need further explanations (ref #1, point #3)

We updated the main text to address these questions. The question about the relative contraction rates is now discussed at lines 295-302. Some discussion about the assumptions and limitation of the theory were added at lines 305-309. All other changes are noted in the specific responses below.

For the simulations shown in the figures or as movies in this paper, we would invite you to include the Cytosim (.cym) files with the respective parameters used to run the depicted simulations. These files can be zipped (ideally with a README file) and uploaded as datasets labeled "Computer simulation file (Cytosim) for Figure XX-ZZ".

We prepared all necessary simulation configuration scripts in separate zip files grouped by figure label, as suggested. The instructions on how to download and run the software from GitHub are included in ReadMe files.

Please relabel the "Supplementary note" into "Appendix". It should be referred as such from the main text. Appendix figures and tables should be called out as "Appendix Fig S1", "Appendix Table S1" from the main text.

We have relabeled the Appendix and all the references to this material.

For the HTML version of your paper, please include the following items:

- three to four 'bullet points' highlighting the main findings of your study

- a short 'blurb' text summarizing in two sentences the study (max. 250 characters)
- a 'thumbnail image' (width=211 x height=157 pixels, Illustrator, PowerPoint, OmniGraffle or jpeg format), which can be used as 'visual title' for the synopsis section of your paper.

The contraction or expansion rates of disordered cytoskeletal networks is predicted based on the properties of the filaments, and the molecular motors and crosslinkers that link them.

- The prediction is calculated analytically for networks made of flexible, semi-flexible (actin) and rigid (microtubule) filaments.
- It explains the combined contribution of crosslinkers and motors in producing contraction of actomyosin systems.
- The theory reveals new conditions to produce contractile or expansile cytoskeletal networks.
- It unifies previously proposed mechanisms of contraction into a common framework.

Please resubmit your revised manuscript online, with a covering letter listing amendments and responses to each point raised by the referees. Please resubmit the paper ****within one month**** and ideally as soon as possible. If we do not receive the revised manuscript within this time period, the file might be closed and any subsequent resubmission would be treated as a new manuscript. Please use the Manuscript Number (above) in all correspondence.

When you resubmit your manuscript, please download our CHECKLIST (link) and include the completed form in your submission. *Please note* that the Author Checklist will be published alongside the paper as part of the transparent process (LINK).

Click on the link below to submit your revised paper.

_____
_____

Kind regards, Thomas

We thank the reviewers for their careful reviews of the manuscript. In blue is a response to each point raised. Changes are also highlighted in the manuscript itself.

Reviewer #1

This is an exciting theoretical study that examines contractility of actin-myosin networks. In fact, the authors consider very general class of polymers contracted by ensembles of various types of motors and crosslinkers. The big question, which pre-occupies the field lately, is whether and how myosin can contract random actin arrays - where does an asymmetry, necessary because myosin contracts antiparallel filament pair overlapping near their minus ends, but expands the pair overlapping near their plus ends, come from.

A number of theories was proposed - some (Lenz/Gardel et al) posit that there is a mechanical asymmetry because expanding pair buckles, other (Oeltz/Mogilner et al) - that the asymmetry is due to polar filament treadmilling, yet other (Lenz/Carlsson et al) - that zippering or rotation in 2D is responsible for this asymmetry. What plagues all these theories is that analytical estimates they produce are not very reliable, while numerical simulations provide limited insight.

The powerful approach of the present study is to consider all possible intersections of a filament with other filaments, all combinations of motors and crosslinks at these intersections, and then use elementary mechanics and probability estimates to derive analytically rules for whether there is a contraction or expansion or oscillations of the networks depending on the conditions.

The analytical results are in a shockingly good agreement with numerical simulations. What this means is that there is a comprehensive phase diagram that for the first time allows us to predict the contractile behavior based on densities, filament flexibilities and length distributions.

This is a great result, broadly applicable and interesting, very systems-bio-like, and appropriate for MSB. The manuscript is written very well.

The comments and questions below are not a criticism, but rather something for the authors to consider:

1) as contraction goes on, many theories predict that with time patterns will form in the network that would worsen the contraction. Presumably, numerical simulations then should be fitting the analytical predictions worse? can this be tested?

We focused on pattern formation in the past for one particular type of connector (Nedelec et al. 1997). In our current work, we observe indeed that for many combinations of connectors, the systems evolve into distinct patterns. We observed some very interesting phenomena, but this data is not shown, as this is far too much material to be treated now. In this submitted article, we intentionally focused on the initial evolution of a random network, for which we are able to calculate the number of intersections, the distance between the intersections, and other characteristics analytically. We only consider the earliest phase of network evolution. We likewise expect the analytical prediction worsen as the network organizes and moves away from a random state, and we certainly intend to test how well the prediction holds in a less random state. We have updated the discussion section (lines 305-309) to make clear that addressing network organization is not the objective of this work.

2) For very dense networks, theory probably misses multiple intersections per filament. What does numerical simulation predicts in this case - worse contraction?

This question is based on a misunderstanding. The theory accounts for all the intersections in the network, and although they are considered pairwise, the calculation is not limited in the number of intersections per filaments. In the networks that were simulated to produce Figure 2C, there are ~90 intersections per filament. This is explicitly stated in the caption of new Fig. S6, which also includes a snapshot of the network.

3) There has to be a better qualitative explanation for pulsatile contraction - looking at the movie and figure I see that the effect is due to transient local filament clusters, but what are the essential feedbacks that always underlie oscillations, and what explains the characteristic time scales.

This is one of a set of really interesting questions that we plan to treat in another article. The issue of pulsatility (and indeed even the precise definition of this term!) can now be explored within this theory in many dimensions. An encompassing and thoughtful analysis will take time and would go beyond the frame of this paper. We included the current simulation more as an outlook.

The basic fact is that to avoid that a contracting network collapses into its center, it needs to be reset in some way. What the simulations show is that filament turnover is a sufficient mechanism to achieve that, but there are other ways to achieve this, and we have not yet explored them. At the moment, we do not have a detailed explanation of what sets the characteristic time scales, but we do observe a correlation between the apparent period of oscillations and the average filament lifetime (Fig5 F).

4) there is some, but very sketchy 2D vs 3D network discussion - would be good to see some semi-quantitative estimates of expected differences.

Again, we included this material simply to show that the theory and simulation allows us, and anyone else interested, to study this question in more depth. Box 1 present our prediction for 1D and 3D systems, based on dimensionality. Extending the analysis to full 3D meshworks, or other network, such as (cytokinetic) rings, will be exciting challenges for us or others in the future.

5) Finally, theories of Lenz, Carlsson, Mogilner, others, have some quantitative predictions - how do those compare to the theory in this study?

It is quite difficult to compare quantitatively the prediction from others, simply because the characteristics of the networks are not the same. There is no established 'standard' network and it seems that everyone has adopted different assumptions. For example, Ronceray, Broedersz and Lenz, (PNAS 2015) provide many interesting predictions, but their network is a regular mesh of triangles. How can we relate their network to the random overlap of filaments that is considered in our study? In PRX Fig. 4, M. Lenz predicts the dominant contractile mechanism as a function of the number of myosin heads composing the minifilaments, but we do not consider minifilaments and only used simple bifunctional motors, which is justified by our ambition to address both actin and microtubules systems. In PRL 2006, Carlsson focused on predicting the magnitude of the contractile force, whereas we predict the contractile rate, in the low force regime. In Physica D 2015, Oelz and Mogilner consider a ring of bundled filaments (the geometry of the cytokinetic ring), and this does not match our assumption that the network should be disorganized. They also implemented filament treadmilling which we did not consider. Comparing these different approaches is an outstanding program for a review article, but we feel that it is beyond the scope of this article.

Reviewer #2

This manuscript presents a framework for understanding if mixtures of motors, cross-linkers, and filaments will be contractile or expansile. This is a significant advance that helps unify a wide variety of previous proposals. I believe that this is very important work that provides great insight into the collective properties of the cytoskeleton.

I have a number of suggestions that may help to improve the manuscript.

MAJOR POINTS:

1) The central equation of the presented theory, Box 1, parts A and B, contains the variable v_i , which is the rate of change of the spacing between two motors/crosslinkers. Given that motors have a force velocity relationship, it is unclear how v_i is determined in analytical calculations. Is

v_i given from the unloaded motor velocity v_m ? If so, does that imply that the motors are under low load, which in turn suggest that this theory may fail for large or heavily crosslinked networks. If not, it is unclear how v_i is determined.

For our analytical prediction, we use the unloaded velocity of the motors, and discard the force-velocity relationship. This is a simplification which is expected to hold whenever the network is under low tension. We have stated this assumption more explicitly on lines 295-297.

I think that this issue may be related to, what was to me, a cryptic statement in the Discussion: "Rather than an absolute contraction rate, this theory predicts the relative contraction rates of two networks when the parameters of the connectors (numbers, types, binding rates, unbinding rates) are different between them." It is unclear to me why the author's theory cannot be used to predict absolute contraction rates, or given that, why it does successfully predicts relative contraction rates. It is very important to explain this issue in more detail.

This was not well explained. We can indeed predict the absolute contraction rate, if all the microscopic quantities are known. However, this may not always be the case, and in particular the filament length may remain unknown. In this case, although the theory cannot predict the absolute contraction rate, it can still predict how changing the motor parameters affects contractility. Thus, depending on the level of knowledge on the system, the prediction can be absolute or relative. Of course, in the simulations we know all the parameters, but we intend the theory to be useful to experimentalists. This is now more clearly explained on lines 297-302.

2) In the author's discussion of networks of semi-flexible polymers, they present the results of an analytical calculation (Figure 2C), without giving the detailed calculation that was used to obtain the plotted line. They says that this aspect of their theory was "Guided by the results of many simulations" without presenting the results of those simulations. The authors really do need to provide more details on this aspect of their theory. They should give a full derivation of their analytical calculations and present the simulations that support the validity of their calculations. For example, they should present simulations in which filament density and persistence length are systematically varied and show the extent to which their analytical theory can or cannot these results.

The appendix now includes a section G describing how we derived the calculation for the curve presented in Fig 2C with hopefully enough details to be understood/reproduced.

MINOR POINTS:

1) At some times the authors focus on the effects of the crosslinkers on the network (as very well illustrated in the first three part of Figure 1E), while at other times the authors focus on the effects of the crosslinkers on the filament (as in the "effect" label on Figures 2A and 3A). This distinction is not clearly explained, which makes the presentation confusing at points.

We are always considering the effect that the configurations have on the rest of the network. This seems to be a misunderstanding caused by our usage of multiple terms. We have now modified the text to be more consistent. The corresponding columns of Figure 3 and 4 were relabeled "Category". We also modified the labels of Figure 1E, to define the categories of configurations used in this work (neutral, static, contractile, expansile).

2) The paper would read better if much of the speculation and possible future directions were moved to the discussion section. For example the points about light-switchable motors on line 197, and the line about anisotropic networks on line 232 seem out of place.

We thank the referee for these suggestions, which have been implemented (lines 287-289 and 417-420).

3) The authors state that "We thus simulated networks with all the combinations of connectors that were contractile on Fig. 4B and varied systematically the filament turnover rate. All displayed pulsatile behavior, confirming the universal nature of the phenomenon (data not shown)." It is difficult for a reader to evaluate claims of universality without more details. The authors need to present these results. How did they vary filament turnover? Over what range of values? How exactly did they determine that these networks were pulsatile?

Filament turnover was implemented in simple terms with random deletion of an existing filament and placement of a new one, thus keeping the number of filaments constant. Quantifying pulsatility is not trivial and we used visual inspection to judge if these systems were pulsatile. We have rewritten parts of this section in a more precise manner. We provide all the configuration files with pulsatile behavior as part of the supplemental material. The interested reader will be able to simulate these systems, and develop his own quantification of pulsatility.

4) Figure 4B is difficult to read, especially in black and white, perhaps a different color set (besides red and green) and a different set of symbols would work better. The difference between inward point arrows and outward pointing arrows is also difficult to notice when the paper is printed.

We thank the referee for pointing this out. We decreased the size of the 'contractile' symbols and increased the size of the 'expansile' symbols, to hopefully improve the readability of this figure.

5) On line 182, the use of the word "bivalent" is confusing. Is it intended to contrast with monovalent or trivalent motors?

The word is intended to mean that we are using a connector with two motor subunits in contrast to connectors composed of one motor linked to a binder. We have updated the text on lines 124-126 to make this clearer. We now use 'bifunctional' throughout the paper in a more consistent manner.

Reviewer #3

In the manuscript entitled "A theory that predicts behavior of disordered cytoskeletal networks" the authors use a combination of theoretical modeling and computer simulations to investigate the dynamical behavior of biological networks comprised of active molecular motors, passive cross-linkers and elongated filaments. Previously, the properties of active networks have been extensively studied using coarse-grained hydrodynamic models. However, such models cannot connect microscopic dynamics of constituent unit to the macroscale behavior. Consequently, this is an important subject area that remains poorly understood. The authors describe a comprehensive study. They propose a simplified theoretical model that predicts how the contraction/extension of an active network will depend on the properties of the constituent units. Computer simulations quantitatively test the theoretical predictions over an impressively wide range of molecular parameters. I believe that the manuscript is of high quality and acceptable for publication. I find it quite surprising how a very simple model describes network dynamics over a wide range of parameters. Overall the manuscript is well written but addressing the following few queries would further clarify and improve it.

1. Can the proposed theoretical model quantitatively describe the rheological properties of passive networks? This might be essential for comparison to experimental data. For example, experiments on the contracting actin-myosin gels (Bendix et. al. Biophysical Journal 2008) show that myosin molecular motors can induce macroscopic contraction only at intermediate cross-linker concentrations. Presumably at high cross-linker concentrations the stiffness of the network is too high for motors to induce network contractility. It is not obvious that the model captures this effect. This is not essential for publication but it would be useful to comment on.

Our theory successfully explains the results by Bendix et al., as the contractile rate indeed decreases as crosslinkers are added beyond a certain level. This is a general feature of the model,

as explained on lines 151-155. For the simulations presented in Figure 2, optimal contractility is obtained when the number of crosslinkers is about equal to the number of motors. This is the case because the binding rates are equal for motors and crosslinkers. The fact that contraction is inhibited by addition of crosslinkers can also be seen in Figure 2D: with 16 motors per filaments (third line from top), contraction at 64 or 32 crosslinkers per filament is significantly slower than with 16 crosslinkers per filament. Our Figure 2D in this respect match the diagram presented on Figure 2D of Bendix et al, Biophysical J. 2008. We have updated the text (lines 190-192 and 321-323) and the panel of Figure 2D to make this important feature of our work more visible.

2. The authors state: "to predict if the whole network will contract or expand, we sum up the effect of all elementary mechanical bridges in the network." I think that the idea that the macroscopic contraction or expansion can be directly related to dynamics of single filaments is intriguing. However, I would like to see more detailed explanation of how one relates contraction of individual segment to the overall network contraction. Furthermore, their model involves a number of assumptions that should clearly be stated in the text. For example, I believe that that the model assumes that the network structure remains the same throughout the entire contraction process. This is not necessarily the case in experiments, as the filaments can form bundles at high crosslinker concentrations or orientationally ordered nematics. How does the model account for such possible dynamical pathways?

We have updated the manuscript to better explain how the prediction is calculated (Box 1). The fundamental assumption that the network is percolated/connected and randomly organized is stated in lines 86-91, and the Appendix includes a detailed description of the assumptions with justifications. Our assumptions exclude any kind of structure such as bundles or nematic islands. Thus, currently the theory does not account for this, but we certainly think that the theory can be extended in this direction. For a truly random network, the relative movement of doubly bound connectors at each filament segment will generate and transmit either a contractile or expansile stress to the rest of the network. We have also added a few sentences about these issues to the discussion section (lines 295-302 and 305-309).

3. The senior author of the submitted manuscript has previously shown in an important advance that clusters of molecular motors and filaments can assemble into mesoscopic structures such as asters and vortices. Can the current model predict formation of such structures? This should be discussed.

Our theory covers the initial response of the network, and should remain valid as long as the network remains nearly isotropic and percolated. With time, networks may reorganize into asters, bundles or other structures, which may be deduced from the molecular players involved in each situation, but this is not within the scope of our submission (see lines 305-309).

4. The model assumes that there are no excluded volume interactions between filaments. In absence of such interactions, what ultimately determines the final density of the contracting network?

Our article only deals with the initial stage of contraction (or expansion). We have not included steric interactions for simplicity, and in the simulation contractile networks may contract all the way to a single spot, if given sufficient time. This is visible for example on Figure 2D, top right corner and Figure 3E, bottom right corner. Networks at these later time points can reach arbitrary density, but we have not attempted to 'fix' this, since studying the long-time evolution of the network is beyond the scope of our study. We selected these time points in the illustrative panels, as they are visually easier to see, but for the more quantitative panels (Figure 2C and 3D) earlier time points were used where the network only contracted a small fraction of its size.

2nd Editorial Decision

31 August 2017

Thank you again for submitting your revised work to Molecular Systems Biology. We are now satisfied with the modifications made and I am pleased to inform you that we will be able to publish your paper in Molecular Systems Biology pending the following minor amendments:

- please reformat the citation/bibliography to the MSB style
- please include the .tpl file in the zip archive for the scripts related to figure 2 and include an explanation in the README file as how to use it (with reference to Preconfig) and in what sense it relates to the files in the folder Fig2C already included in the package.
- please include a short running title
- please include a conflict of interest statement (even if there is no COI)
- please include a set of keywords in the word file of your manuscript.
- to be able to render correctly your paper in HTML, Dataset EV1, Movie EV1, EV2 EV3 and EV4 need to be explicitly called out from the main text. Please include these.
- please note that figure 2B is never called out explicitly and check whether this is intentional or an omission.
- please reduce the size of the synopsis image to a width of 550 pixel width maximum and a height of 400 pixels maximum

2nd Revision - authors' response

31 August 2017

Authors made the requested changes and submitted the final version of their manuscript.

3rd Editorial Decision

05 September 2017

Thank you again for sending us your revised manuscript. We are now satisfied with the modifications made and I am pleased to inform you that your paper has been accepted for publication.

Corresponding Author Name: Francois J. Nedelec

Manuscript Number: MSB-17-7796